# Tissue-Resident Memory T Cells in the Liver—Unique Characteristics of Local Specialists

**DOI:** 10.3390/cells9112457

**Published:** 2020-11-11

**Authors:** Lea M. Bartsch, Marcos P. S. Damasio, Sonu Subudhi, Hannah K. Drescher

**Affiliations:** Division of Gastroenterology, Massachusetts General Hospital and Harvard Medical School, Boston, MA 02114, USA; mdamasio@mgh.harvard.edu (M.P.S.D.); ssubudhi@mgh.harvard.edu (S.S.)

**Keywords:** liver, tissue-resident memory T cells (T_RM_ cell), HBV, HCV, NAFLD

## Abstract

T cells play an important role to build up an effective immune response and are essential in the eradication of pathogens. To establish a long-lasting protection even after a re-challenge with the same pathogen, some T cells differentiate into memory T cells. Recently, a certain subpopulation of memory T cells at different tissue-sites of infection was detected—tissue-resident memory T cells (T_RM_ cells). These cells can patrol in the tissue in order to encounter their cognate antigen to establish an effective protection against secondary infection. The liver as an immunogenic organ is exposed to a variety of pathogens entering the liver through the systemic blood circulation or via the portal vein from the gut. It could be shown that intrahepatic T_RM_ cells can reside within the liver tissue for several years. Interestingly, hepatic T_RM_ cell differentiation requires a distinct cytokine milieu. In addition, T_RM_ cells express specific surface markers and transcription factors, which allow their identification delimited from their circulating counterparts. It could be demonstrated that liver T_RM_ cells play a particular role in many liver diseases such as hepatitis B and C infection, non-alcoholic fatty liver disease and even play a role in the development of hepatocellular carcinoma and in building long-lasting immune responses after vaccination. A better understanding of intrahepatic T_RM_ cells is critical to understand the pathophysiology of many liver diseases and to identify new potential drug targets for the development of novel treatment strategies.

## 1. Introduction

T cells play a central role in the immune response against pathogens. CD8+ T cells are highly effective in the eradication of cells infected with pathogens, damaged cells and even cancer cells. CD4+ T cells contribute to and modulate the immune response against pathogens and in the cancer environment. A very important characteristic of the adaptive immune system is to build up a pool of memory T cells, which enables a fast and effective immune response after pathogen re-challenge. These memory T cells patrol in the circulation in order to encounter a known pathogen. In addition to the blood circulation and lymphoid tissues, memory T cells can also be found in non-lymphoid tissues. Within these tissues, they mediate a fast induction of the immune response after the second pathogen challenge and initiate the recruitment of other immune cells by modifying the tissue-specific inflammatory microenvironment. Different scientists were able to detect a persistent tissue-resident memory CD8 and CD4 T cell subpopulation at different tissue sites. These cells can be mainly found in organs that are frequently exposed to pathogens, such as the liver, skin, gut and lung [1]. These resident memory T cells (T_RM_ cells) are known to be important for pathogen surveillance in the respective tissue and have a distinct phenotype in comparison to their circulating counterparts in the blood [2,3]. Interestingly, the tissue itself influences the phenotype of the memory T cell. This review will focus on the development and functional role of CD8+ T_RM_ cells, especially in the liver, and concentrates on the characteristics that make them unique compared to other T_RM_ subpopulations. The liver is an important immunological organ as it is not only affected by many different hepatotropic viruses but is also the first control point of pathogens that enter or re-enter the body through the gastrointestinal tract. Liver tissue contains a diversity of immune cells and interestingly has a special composition of these immune cells in comparison to other tissues or the blood. In order to prevent a systemic infection, the liver plays an important role as a gatekeeper and an effective T_RM_ cell population contributes to effective pathogen clearance.

## 2. Phenotype and Development of T_RM_ Cells in the Liver

Memory T cell development is an important characteristic of the adaptive immune system and critical for an effective immune response after a second challenge with a pathogen. In order to build systemic protection, circulating and tissue-resident memory T (T_RM_) cells exist. Based on their expression of cell surface markers and specific functions, circulating memory T cells can be further divided in different subpopulations: central memory T cells (T_CM_) and effector memory cells (T_EM_). These memory T cells typically circulate through the body in order to encounter pathogens but were also found in lymphoid tissues. In contrast, T_RM_ cells have the ability to persist within a certain tissue.

### 2.1. T_RM_ Cell Phenotype

The first detection of T_RM_ cells was possible by analyzing the expression of cell surface markers on a specific memory T cell pool within the tissue. These markers are described to maintain their tissue-specific function. Liver T_RM_ cells usually downregulate the expression of homing receptors such as CCR7 and CD62L. Furthermore, they decrease the expression of tissue egression markers, such as soingosine-1-phosphate 1 (S1PR1), while upregulating CD69. Additionally, liver T_RM_ cells express other adhesions and functional molecules, such as CD103, CD49a and CD44, to enable trafficking within the liver sinusoids and liver tissue [4]. CD69 expression is strongly initiated after T cell activation. Besides being an activation marker, it also mediates T cell homeostasis and cell migration–retention [5]. CD49a is the α1 subunit of the α1β1 integrin, also known as very late antigen-1 (VLA-1). It binds to collagen IV and mediates T cell adhesion. It is therefore important to establish tissue residency [6]. CD44 is another adhesion molecule that is upregulated after T cell receptor (TCR) stimulation and promotes T cell migration [7]. CD103 is the alpha E subunit of the alpha E beta 7 integrin, an adhesion molecule defined as a marker of T_RM_ cells, especially in human tissues [8]. In the liver, on the other hand, CD103 is not expressed in all T_RM_ cells. The CD69+CD103− subpopulation is proposed to be a subpopulation that can recirculate and take on other functions upon pathogen challenge, whereas the CD69+CD103+ are defined to be truly liver resident [9]. In contrast, it could be shown that the majority of CD4+ T_RM_ cells are long living and tissue-resident cells that express high levels of CD69 and show a low CD103 expression [10].

The effector function of T_RM_ requires constant chemokine stimulation. Therefore, liver T_RM_ cells constitutively express the chemokine receptors C-X-C motif chemokine receptor 3 (CXCR3) and C-X-C motif chemokine receptor 6 (CXCR6) on their surface. CXCR3 is an important homing marker and supports the maintenance of T_RM_ cells in the liver tissue. CXCR3 has the ability to bind multiple ligands predominantly secreted by monocytes, endothelial cells and fibroblasts, e.g., chemokines C-X-C ligand 9 (CXCL9), C-X-C ligand 10 (CXCL10) and C-X-C ligand 11 (CXCL11) [11]. CXCR6, on the other hand, is essential for T_RM_ cell development but also supports the maintenance of T_RM_ cells in the liver through binding C-X-C ligand 16 (CXCL16) secreted by liver endothelial cells [12] (Figure 1).

In addition to the surface markers, transcriptional analysis of T_RM_ cells showed that they express a distinct transcription factor profile [13]. T_RM_ cells, including liver T_RM_ cells, upregulate the transcription factors homolog of blimp-1 (HOBIT) and BLIMP1 [14,15]. HOBIT was first described to be upregulated in CD45+ effector T cells after viral infection, e.g., human cytomegaly virus infection (CMV). In T_RM_ cells, HOBIT and BLIMP1 actively downregulate the expression of C-C chemokine receptor 7 (CCR7), Kruppel-like factor 2 (KLF2) and sphingosine-1 phosphate receptor 1 (S1PR1) [16]. CCR7 is highly expressed in naïve cells and T_CM_. It is the receptor of chemokine ligand 19 (CCL19) and chemokine ligand 21 (CCL21) that convey migration of cells to secondary lymphoid tissues [17]. Opposingly, KLF2 regulates the expression of S1PR1, which directs cells from a specific tissue into the periphery [18]. Therefore, the co-expression of HOBIT and BLIMP1 represses the formation of circulating memory T cells and silences the genes related to cell recirculation into the periphery. In a recent mouse study, Park et al. demonstrated that HOBIT expression in liver T_RM_ cells is mediated by the gene repressor *Capicua* in collaboration with the ETS variant transcription factor 5 (ETV5), underlining the importance of this transcription factor [19]. In addition to HOBIT and BLIMP1, the transcription factors Runt-related transcription factor 3 (RUNX3), TBX21 (Tbet) and Notch were reported to be upregulated in T_RM_ cells after their development and their expression is essential for a sustained T_RM_ cell population [20]. RUNX3 represses the expression of genes involved in the activation of circulating memory T cells. On the other hand, RUNX3 induces the expression of genes such as integrin subunit alpha E (ITGAE), which encodes for CD103 in T_RM_ cells and mediates the production of granzyme B by T_RM_ cells [20].

Tbet is known to mediate the expression of the IL-15 receptor (IL-15R) in order to establish a long-term lineage stability. The membrane bound transcription factor Notch is predominantly expressed in newly developed T_RM_ cells and responsible for their maintenance through the regulation of their metabolic profile [20] (Figure 1).

### 2.2. T_RM_ Cell Development

The rather recent identification of T_RM_ cells as a distinct tissue-resident memory T cell subpopulation leads to the question: Which specific factors are involved in their development and maintenance, and which are crucial for their tissue-specific function?

Different models exist to explain the origin and development of memory T cells after a pathogen challenge but the overall development is not fully understood yet [21,22]. It is still unclear whether T_RM_ cells and circulating memory T cells originate from the same precursor cell subset. It is further unclear whether liver T_RM_ cells develop extrahepatic and migrate into the liver or whether they directly differentiate intrahepatic. Adoptive transfer experiments in mice demonstrated that in vitro-activated CD8+ cells can differentiate into T_RM_ cells after transfer into the specific tissue and are not further distinguishable from those generated within the tissue itself [23]. Based on the current literature, which will be discussed in more detail in this review, we presume that both extra- and intrahepatic development contributes to the T_RM_ pool in the liver.

Several factors are known to contribute to T cell development in general and further determine the specificity of a memory T cell and the fate of T_RM_ cells.

#### 2.2.1. Origin of T_RM_ Cells

To answer the question about T cell origin, one important approach is to analyze the TCR repertoire of the cells. In search of the origin of T_RM_ cells different groups analyzed their TCR repertoire and could find that T_RM_ and T_EM_ have an overlapping TCR repertoire, suggesting that these subsets develop from the same progenitor cell [24]. Holz et al. were the first to describe that liver T_RM_ cells also require TCR stimulation and rearrangement upon binding to a specific antigen for their formation [25].

In addition, TCRs differ in their strength of antigen binding, which influences the development of effective CD8 memory T cells. A low affinity TCR stimulation leads to insufficient memory T cell development of cells with a short lifespan and thus leads to impaired secondary immune responses. Furthermore, the strength of TCR binding varies between different memory subsets, e.g., the binding strength has to be especially high for T_EM_ development. However, the specific strength of TCR stimulation for the development of liver T_RM_ cells has not been described yet but the described overlap with the TCR repertoire of T_EM_ cells indicates the potential requirement of high-affinity binding [8].

Another important marker to investigate the origin of T_RM_ cells is the killer cell lectin-like receptor G1 (KLRG1), which is usually upregulated in effector T cells and relatively low expressed in memory precursors of circulating memory T cells.

Mouse studies could demonstrate that approximately 50% of the intrahepatic T_RM_ cell population develops from a KLRG1-expressing precursor whereas the other 50% develop from precursors that only transiently express KLRG1 [26]. These findings indicate that intrahepatic T_RM_ cells develop from two different origins. Cells simultaneously develop extrahepatic from memory precursors and migrate into the liver and another proportion directly develops within the liver from effector T cells. T_RM_ cell tissue dependency might even be less strict as it was recently shown that a certain subset of T_RM_ cells can leave their resident tissue and re-circulate into the blood (Figure 2). These previous T_RM_ cells expresses a similar transcription factor repertoire and surface molecule pattern as their tissue-resident counterparts.

These findings suggest that although T_RM_ cells might be able to reside in a specific tissue for a long period of time, they are able to recirculate if needed [27]. A recently published study by Pallett et al. could nicely demonstrate the longevity of human liver T_RM_ cells. They could demonstrate that, in liver allograft tissue, 2–6% of CD8+ T cells have a donor-derived T_RM_ phenotype and were detectable 11 years post transplantation. Additionally, they showed that recipient CD8+ and CD4+ T cells could develop a T_RM_ phenotype although the recipient-derived T_RM_ cells tended to express fewer liver-resident markers. Furthermore, a small population of donor-derived CD4+ T_RM_ cells was detectable [10]. To conclude, there are strong indications that liver T_RM_ cells can develop from two different origins (Figure 2).

#### 2.2.2. Factors Influencing/Driving T_RM_ Cell Development

Within the tissue, T_RM_ cells can be exposed to a variety of environmental conditions that influence their development, maintenance and function, e.g., nutrient deprivation, hypoxia and the inflammatory cytokine milieu [20].

Certain cytokines are described to promote T_RM_ cells differentiation. One important cytokine is the tumor growth factor β (TGF-β), which induces the expression of CD103 on the cell surface [28]. Furthermore, monocyte-derived IL-10 facilitates the TGF-β release and promotes CD103 expression on T_RM_ cells. Blocking of IL-10 decreased the CD103 expression [29].

In addition, interleukin-7 (IL-7) and IL-15 are involved in T_RM_ development and longevity [25]. The importance of these cytokines is shown by the upregulation of the IL-7 and IL-15 receptors in T_RM_ cells. Thus, it could be shown that IL-15 stimulation is especially essential for hepatic T_RM_ development as IL-15^−/−^ mice were prevented from developing these cells in the liver [25]. Interestingly, the requirement to the cytokine microenvironment differs between CD4 and CD8 T_RM_ cells. CD8 T_RM_ cells require IL-15 and IL-7 stimulation, whereas CD4+ T_RM_ stability only depends on IL-7 signaling [30].

In addition, the pro-inflammatory cytokines type 1 Interferon (IFN) and IL-12 are described to positively influence T_RM_ cell development by inducing the expression of CD103 and CD69 [27].

Another cytokine that functions as an autocrine stimulus on CD8 T_RM_ cells is IL-2. Pallett et al. showed that human liver CD8 T_RM_ cells produce high levels of interleukin-2 (IL-2), which was shown to be critical for intrahepatic T_RM_ cell survival, the ability to function and for their antigen-specific proliferation [11]. Increased IL-2 expression could also be shown in response to oxidative stress induced by stimulation with reactive oxygen species and free radicals, which is a common characteristic of chronic liver diseases such as non-alcoholic steatohepatitis (NASH) [31]. Thus, oxidative stress induction could alter T_RM_ cell function and differentiation.

Comparable to many other T cell subsets, also the metabolic profile influences T_RM_ cell development, maintenance and function. Different T cell subsets preferentially use a certain metabolic pathway to gain their energy. In general, highly proliferating and active cells favor glycolytic pathways, whereas quiescent cells mainly use oxidative phosphorylation and FAO to generate ATP. A key regulator of T cell metabolism, proliferation, activation and survival is the transcription factor mammalian target of rapamycin (mTOR) [32]. mTOR consists of two subunits, mTOR complex 1 (mTORC1) and mTORC2. Both complexes can be activated in T cells by various stimuli like TCR stimulation, cytokine ligation and the presence or absence of certain metabolites. mTOR activation in turn leads to a downregulation of S1P1 and KLF2 in activated T cells.

mTOR can further induce glucose consumption to support T cell proliferation and thereby favors the development and tissue retention of T_RM_ cells [33]. The importance of mTOR signaling in T_RM_ cells could be demonstrated by using rapamycin. Rapamycin is an mTORC1 inhibitor that was shown to reduce T_RM_ formation in mucosal tissues by inhibiting CD103 and CCR9 upregulation in T_RM_ cells and cell migration. The effect of rapamycin was tissue-specific for the gut-associated lymphoid tissue and the exact implications on the liver are still under investigation [34].

More important and better investigated is the impact of fatty acid oxidation (FAO) on T_RM_ cells. It could be demonstrated that T_RM_ cells rely on FAO to gain their energy. In order to take fatty acids up, T_RM_ cells upregulate fatty acid-binding proteins (FABP). T_RM_ cells from different tissues express distinct FABPs with varying fatty acid specificity, depending on the specific tissue of origin [35]. It could be demonstrated that liver T_RM_ cells express mainly FABP1 and FABP4 in a low concentration but do not express FABP5. Thereby, the expression of specific FABPs are upregulated during T_RM_ development and maturation in a tissue-dependent manner to optimize the usage of local fatty acids. Frizzell et al. were able to stress the importance of FAO in a mouse model of LCMV infection. FABP1 deficiency in infected mice was associated with impaired liver T_RM_ cell development [36].

Additionally, liver T_RM_ cells express high levels of P2X purinreceptor 7 (P2RX7). P2RX7 is a sensor for extracellular ATP (eATP) and is involved in various inflammatory processes [37]. Studies suggest that P2RX7 promotes oxidative phosphorylation and FAO and thereby regulates the metabolic function of CD8 T_RM_ cells. P2RX7 deficiency, on the other hand, prevents a stable and durable T_RM_ cell development and reduces the expression of anti-apoptotic molecules. Furthermore, P2RX7 expression was shown to positively influence IL-7 secretion and IL-15 receptor expression and thereby influences T_RM_ cell development [37,38].

Another characteristic of the liver is the presence of hypoxic regions due to its venous blood supply from the portal vein in combination with slow blood flow in the sinusoids. Interestingly, T cells have the ability to adjust the expression of the hypoxia inducible factor 1α (HIF-1α) and HIF-2α as they detect the local oxygen gradient [39]. HIF-1α and HIF-2α are transcription factors that play a crucial role in the cellular response to low oxygen concentration, especially T cell development, metabolism and function.

The finding that an intrahepatic subpopulation of T_RM_ cells can be found that expresses CD69+CD103− and HIF2α suggests that they are predominantly located in hypoxic regions within the liver. Interestingly, this T_RM_ cell subpopulation could not be located in other tissues such as the lung, skin or colon [40]. This finding underlines the importance of tissue-specific adaptions of T_RM_ cells to unique environmental conditions in different organs.

## 3. The Phenotype and Transcriptional Profile of Liver T_RM_ Cells in Mouse and Man

Blood is the major human sample type for the study of T cell immunology because it is easily available. Human tissue samples, especially liver tissue, can just be obtained from living individuals most often suffering from end-stage liver disease and is limited to invasive tissue sampling through surgical resection, biopsy or more recently fine needle aspirates (FNA). This lack of sample disposability lead to extensive investigations of liver T_RM_ in different rodent models. Various in vivo studies, and not only on their differentiation, maintenance and function but also on their phenotypic and transcriptional features, have been conducted in infection mouse models such as hepatitis B virus (HBV) or malaria.

T_RM_ are phenotypically characterized by their expression of CD69 ± CD103. Transcriptional analysis of these cells in different human tissues, such as skin, lung and liver, revealed that they exhibit a distinct profile, discriminating them from circulating memory T cells. This profile is also mirrored in its key features in mouse T_RM_ cells, making them a suitable model for the investigation of these cells in health, disease and autoimmunity. However, the translational potential of animal models, especially in the course of inflammatory disorders, is still under debate. Although they are important to investigate basic mechanisms, points of criticism are especially that animal models do only partially mimic the complex features and length of chronic injury as they naturally occur in humans. Furthermore, the variety of exposures to different pathogens and environmental influences over the whole disease period hardly can be reflected in any rodent model. Nevertheless, the impact of how much these differences affect the transferability of the basic findings obtained in rodent models, on general and tissue immunity in men, is still unknown. To overcome some doubts, recent studies were performed with so called outbred or dirty mice purchased from pet stores. The commonly used inbred mouse strains are used for basic research due to their comparability based on the removal of genetic variability. Outbred mice in comparison show a higher genetic and phenotypic diversity, which does better parallel or even exceed the human variability. Therefore, these animals have been lately used in different disease models, e.g., when investigating the development and tissue specificity of memory T cells.

It was found that these dirty mice better reflect the human situation especially with regard to homeostatic phenotypical manifestations of T_RM_ cells and may therefore be a conceivable addition and validation to the typically used inbred mouse models. As the microbiome is supposed to have a particular impact on T_RM_ cell development, genetic and phenotypic variability is especially important to investigate this subpopulation. Anyhow, it is yet unclear to what extent these models mirror the human immune reaction in disease [41,42].

### 3.1. Experimental Models

T_RM_ cells are characterized by their continued presence at a specific tissue site, independent from their circulating counterparts. A variety of experimental animal models was used not only to define the phenotypic manifestations of tissue residency but also to determine the potential of these cells to recirculate into the periphery. Mouse models include parabiosis, the direct in vivo labeling of cells with different antibodies; T cell depletion, the transplantation of specific tissues; and direct profiling.

Parabiotic surgery describes the conjunction of the blood circulation of two different mice. One animal got previously infected to induce the development of T_RM_ cells, whereas the other animal is pathogen naive. In addition, animals are distinguished by their genetic CD45+ cell isoform (CD45.1 and CD45.2). This allows to investigate the circulation and migration potential of T cells and to define the phenotypic and transcriptional characteristics of T_RM_ cells between parabionts. Circulating blood T cells usually attain homeostasis between parabionts within one week. T_RM_ cells on the other hand do not exchange between the donor and recipient and reside in the specific tissue of each animal. This technique made it possible to define and investigate T_RM_ cells as a distinct cell population. However, results obtained in parabiosis have to be critically evaluated as the inflammation caused as a consequence of the surgery itself may recruit circulating T_RM_ cells from the specific tissue sites, blurring the differences between the tissue-resident and circulating T_RM_ cell subsets [43,44,45].

The in vivo labeling of circulating cells with fluorochrome-conjugated or depleting antibodies, such as anti-CD90/Thy-1, is another approach to distinguish circulating from tissue-resident T cells to investigate T_RM_ cells in mice. CD90/Thy-1 is expressed on peripheral T cells, thymocytes and is widely used for the depletion of T lymphocytes. Via intravenous injection of these antibodies all circulating cells get labeled or depleted whereas those residing in a specific tissue niche stay unaffected. This allows a really fast and easy localization and investigation of T_RM_ cells in general, although there are some disadvantages when studying T_RM_ cells of the bone-marrow and in liver sinusoids as the bone-marrow compartment is not targeted by depleting antibodies [46,47]. Especially in the liver, reports suggest that the sinusoids contain a fraction of sinusoidal resident CD8+ T cells, which are affected by the antibody depletion applied to the vasculature but have slightly different features from other liver T_RM_ cells [48,49]. A related technique is the treatment with FTY720.

FTY720 is a sphingosine 1-phpsphate receptor 1 agonist leading to peripheral lymphopenia and preventing the transmigration of T_RM_ cells back into the circulation [50]. This allows the investigation of liver specific T_RM_ cells as they are retained within the tissue while at the same time a systemic lymphopenia is induced.

Additionally, solid organ transplantation is another way to study T_RM_. Mouse experiments transplanting the intestine or parts of the skin gave interesting insights into the longevity, residency and recirculation potential of T_RM_ [3,51].

Most of the presented techniques are not applicable in humans, making the investigation of the phenotypic and transcriptional appearance of T_RM_ cells difficult. However, few clinical approaches such as T cell depletion therapy and organ transplantation allow us to study T_RM_ cells in human tissues. Anti-CD52 treatment, for example, is commonly used as therapy in T cell lymphoma and leads to the targeted elimination of circulating T cells without affecting tissue-resident CD4+ and CD8+ T cells [51]. A very recent study from Pallett et al. was able to investigate HLA-mismatched liver transplant patients, allowing to discriminate liver-resident (donor) from newly infiltrating (recipient) T cells. Allogenic liver tissues were rapidly infiltrated by recipient T cells that underwent reprogramming to express the T_RM_ markers CD69, CD103 and CXCR3^hi^, which is important for intrahepatic retention [10]. Interestingly, long-lived liver-resident donor T_RM_ cells could even be detected after more than a decade (2–6% of intrahepatic CD8+ T cells) and it was found that they do not egress into the recipient circulation via the hepatic vein. This long-lasting tissue maintenance could be also shown for T_RM_ in other tissues. In mice, T_RM_ cells are maintained for several months in tissues such as the skin, intestine, lung and brain. Human T_RM_ cells, in contrast, are maintained in their tissue-specific niche for several years [52]. Some T_RM_ subpopulations were even found to be maintained over an entire life span [53,54]. Comparable findings of tissue-specific longevity could be shown for human intrahepatic NK cells while T_RM_ cells in the intestine and lung were described to be maintained in the tissue for at least 1 year after transplantation [55,56,57].

### 3.2. The Phenotypical and Transcriptional Differences of Liver T_RM_ in Mouse and Man

The description of tissue-specific T_RM_ as a distinct T cell subset started about 10–15 years ago. Since then, groups have tried to distinguish these T_RM_ from their circulating counterparts in both humans and mice. The main marker to define these cells is CD69, which was originally described as marker for early T cell activation. However, it is constitutively expressed by almost all liver-resident CD4+ and CD8+ memory T cells across species and serves as a canonical signal for tissue retention [18]. Interestingly, the depletion of CD69 in mice showed an alternative, CD69-independent way of T_RM_ formation.

Animals indeed displayed significantly reduced CD8+ T_RM_ in skin and lung but CD4+ T_RM_ formation was not affected. Given the fact that the liver generally has a larger amount of CD8+ than CD4^+^ T cells it seems likely that the majority of liver T_RM_ are defined by their expression of CD69.

The most interesting and striking difference between hepatic T_RM_ in the mouse and human is the expression of CD103. Although it is expressed by different subsets of mouse and human CD8+ T_RM_, e.g., in tissues at barrier sites [58], CD103 is not expressed in intrahepatic CD8+ T_RM_ in mice while human liver CD8+ T_RM_ can be either CD69+/CD103+ or CD69+/CD103− [59]. On the other hand, both mouse and human CD8+ T_RM_ express CXCR6, a chemokine receptor that binds CXCL16. It serves as another important marker promoting the establishment for both human and mouse CD8+ T_RM_, especially in the skin and the liver (Figure 3) [12,60]. In mice, the transcriptional landscape of liver T_RM_ gets more and more defined and understood. McNamara and colleagues were further able to show in more detail what molecular interactions of adhesion molecules are responsible to retain CD8+ T cells the liver tissue but at the same time allow their special patrolling function within the liver sinusoids. The initial migration of CD8 T cells into the liver was found not to be superficially mediated by the binding of selectins but rather mediated by interactions with platelets that bind to endothelial cells via CD44 [61]. Recent studies proposed that the intracellular adhesion molecule 1 (ICAM-1)/lymphocyte function associated antigen 1 (LFA-1)-axis is required for the retention of naïve and activated CD8+ T cells and NKT cells in the liver upon antigen presentation [62,63]. With intra-vital imaging it could be shown that ICAM-1/LFA-1 interactions are crucial for CD8+ T_RM_ cells to move along the liver sinusoids and within liver tissue. Hence, the depletion of LFA-1 in mice lead to the disability to develop intrahepatic residence. It seems that, in mice, LFA-1 but not CD103 is responsible for the retention of intrahepatic T_RM_ cells and stresses that the distinct expression of different adhesion molecules is crucial for T_RM_ cells to patrol within specific tissues rather than at barrier sites [64].

However, the transcription factors that drive the formation of T_RM_ in humans remain not well defined yet and is a topic of current investigations. In this context, Pallett et al. were able to identify IL-2 as a marker that is highly expressed by long-lived liver-resident CD8+ T_RM_ cells. They could further show that this holds true for either global or virus-specific intrahepatic T_RM_ [11].

When comparing the compartmentalization and tissue niches of T_RM_ in mice and humans, another difference can be found. While in mice the non-circulating liver T_RM_ account for 40–60% of the liver-resident T cells, this amount is significantly higher in humans, where it ranges between 60 and 80%.

## 4. Liver T_RM_ Cells—In Health, Disease and Vaccination

Memory T cells play a fundamental role in the immune response against a recurring pathogen but are also described to be crucial in cancer immunology and autoimmunity. The overall understanding of T_RM_ cells is still under investigation. A significant characteristic of T_RM_ cells is that they can remain in a distinct non-lymphoid tissue without recirculating into the blood [3,65]. To perform their particular function—to clear recurring pathogens and quickly initiate a confined immune response—the location in the tissue is of special importance. Usually, T_RM_ cells are hardly found intravascular, not even in highly vascularized organs. One important exception is the liver, where one fraction of T_RM_ cells can be found in the vascular space of liver sinusoids [46]. The fenestrated epithelial layer, which is a unique characteristic of the liver, allows a direct interaction of T_RM_ cells with hepatocytes [47,66]. Although T_RM_ cells are found intravascular, the main fraction persists within the liver tissue. In order to fulfill their border patrol function, migration within the tissue is important to encounter the cognate antigen and requires a close interaction with hepatocytes and other liver-resident immune cells. The average speed of the liver T_RM_ migration is relatively fast in comparison to other T_RM_ cells in other non-lymphoid tissues with an average speed of 5–7 µM/min, representing their efficiency and activity. Thus, in comparison to skin-resident T_RM_ cells, liver T_RMs_ are very motile. The average speed of their migration in comparison to non-tissue-resident lymphocytes, on the other hand, is very slow because T_RM_ cells stop repeatedly to establish an immunological synapse. To conclude, the slow kinesis of T_RM_ cells enables the local immune-surveillance and promotes their function [58].

In case of a re-infection, circulating memory T cells contribute to the effective immune response against the pathogen. The re-stimulation in the draining lymphoid organs and migration of these cells into the liver takes several days and mainly promotes T_CM_ cell development. In contrast, circulating T_EM_ cells are recruited into the liver by proinflammatory signals within hours. Thus, the initiation and direct response is created by T_RM_ cells and is essential to avoid pathogen spread and a systemic infection. After encountering their cognate antigen within the tissue, hepatic T_RM_ cells need to rapidly proliferate and provide an effector function in order to resolve the re-infection. After activation, CD8+ T_RM_ cells produce IFN-γ and TNF-α and have the ability to directly lyse target cells. The pro-inflammatory cytokine expression on T_RM_ cells is elevated in comparison to the circulating memory T cells, indicating their efficient effector function at the tissue-site of infection. Additionally, CD8+ T_RM_ cells recruit other immune cells by chemokine production after antigen recognition [67].

They express high levels of CCL3, CCL4 and CXCL1 and can induce the chemokines CXCL9 and CXCL10 even in an IFN-γ-dependent manner [15]. These chemokines contribute to the migration and expansion of neutrophils and monocytes in the liver [68,69]. In addition to the local inflammation, T_RM_ cells can induce a systemic immune response. This could be demonstrated in vaccine studies, which demonstrated a systemic immune response signaled through pro-inflammatory chemokines produced by T_RM_ cells [70]. Schenkel et al. could display that T_RM_ cell quantity increases in the draining lymph nodes after re-infection, which indicates a support of the systemic immune response [71].

In order to efficiently protect against pathogens, liver T_RM_ cells express high IL-2 levels, which allows them to quickly expand after an antigen challenge and maintain their proliferation in homeostatic conditions [11,72]. In comparison to CD8+ T_RM_ cells, the frequency of human CD4+ T_RM_ cells in the liver is relatively low, also potentially due to a reversed CD8/CD4 ratio compared to the blood [73].

Other resident T cells were shown to complement T_RM_ function in the course of pathogen infection. One of the subpopulations are γδ T cells. Usually, the TCR receptor consists of an α and β subunits but a small proportion of T cells express a γδ TCR. These 3–5% of the intrahepatic lymphocytes are called γδ T cells. Although the concrete TCR recognition mechanism is not fully elucidated yet, γδ T cells mostly recognize lipid antigens in the liver, which are presented by hepatocytes. Furthermore, they can sense extracellular stress factors by multiple receptors, e.g., Toll-like receptors (TLR). After activation, γδ T cells produce a broad variety of pro-inflammatory cytokines, such as IFN-γ, TNF-α and IL-17, and chemokines, e.g., RANTES, IP-10 and lymphotactin. Additionally, they express perforin, granzyme and TRAIL and can thereby directly cytolyze infected cells. After a pathogen challenge in the liver, the γδ T cells mainly produce IL-17 [74]. Additionally, the CD8αα γδ T cells exist and are liver-resident cells. CD8αα binds to the class 1 MHC molecule H2-Q10 expressed on hepatocytes and thereby controls their activation and development [75]. In conclusion, γδ T cells are another liver-resident T cell subpopulation that support T_RM_ cell function in the course of pathogen infection.

In addition, natural killer cells (NK cells) and NKT cells contribute to the pro-inflammatory immune response against pathogens and against cancer cells within the liver. NKT cells are a cell population that combines T cell and NK-cell markers and functions. NK cell and NKT cell subsets are enriched in liver sinusoids and complement the immune surveillances function against pathogens and toxins that reach the liver mainly through the portal vein from the gut [76]. Interestingly, liver-resident NK and NKT cells display a higher function and cytotoxicity in comparison to their circulating counterparts [77]. Similar to T_RM_ cells, NK and NKT cells play a role in chronic liver disease, such as liver fibrosis, hepatocellular carcinoma (HCC) and viral hepatitis. In some diseases, they complement T_RM_ cell function, whereas in others they have an opposing function. One challenge is to analyze NKT cells in non-lymphoid tissues. Inhibition of P2RX7 signaling could be a strategy to restore their function after tissue isolation in order to quantify their frequency in healthy and infected liver tissues [78]. By using this strategy, the investigation of the importance and function of the NK-T cells could be improved.

The interaction of the innate and adaptive immune system in various diseases is tremendously important. It could be shown that modern HCC treatment strategies upregulate the Toll-like receptor (TLR) signaling pathways [79]. This general immune activation could be the link to T_RM_ cell development in this treatment strategy by the secretion of cytokines such as IL-10 [29]. In vaccine studies in the lung against influenza viruses, adjuvants, which are TLR agonist, induce an effective polyfunctional T cell immunity. Nevertheless, the induction of CD69 and CD103 expression was similar between different adjuvants [80]. A direct activation and differentiation of T_RM_ cells by TLR activation could not be shown in the liver yet. It is known that TLR activation occurs in the liver subsequent to the leaky gut syndrome in several chronic inflammatory diseases. Therefore, the investigation of alterations in T_RM_ differentiation and function by TLR signaling is a very important topic for future investigations.

### 4.1. Liver-Resident T Cells in Viral Infection

The course of hepatic virus infection can either be acute or progress to a chronic condition. Chronic infection with hepatotropic viruses can cause tissue damage and lead to liver cirrhosis, liver failure and the development of HCC. Chronic viral infection by the hepatitis B (HBV) and hepatitis C (HCV) virus is one of the main causes of HCC development together with ASH and NASH. Liver T_RM_ cells play a major role in promoting a sufficient antiviral response during viral infection and are especially important during chronic infection. CD8+ T_RM_ cells control the viral replication and generate long-lasting viral protection. CD8+ T_RM_ cells can persist in the liver several years after primary infection and can be enriched in chronically infected patients. Depletion of liver CD8+ T_RM_ cells in HCV re-infection mouse models prolonged the virus persistence and prevented effective viral clearance. Furthermore, recovery of liver CD8+ T_RM_ in the same model lead to virus eradication [81]. In human studies, it could be shown that T_RM_ cells are highly increased in chronic HCV-infected patients and these CD69+ cells have a non-naïve and effector memory phenotype. Likewise, these cells express a specific activation and functional phenotype and are important in controlling chronic HCV infection [82].

The importance of virus-specific T_RM_ cells is even more investigated in HBV infection. In chronic HBV infection, a specific liver CD8+ T_RM_ cell subpopulation could be specified. It could be shown that their abundance was negatively correlated with the virus titer. In addition, T_RM_ cells are enriched in HBV patients who reached viral control in comparison to healthy patients—by having the same frequencies of total T cells in the liver [81]. That might indicate that CD8+ T_RM_ cells contribute to a functional cure for HBV infection. A functional cure in HBV means that the continuous transcription of persistent virus DNA—cccDNA—is controlled by the immune system. Furthermore, virus-specific T_RM_ cells were still detectable in spontaneously recovered HBV-infected patients, stressing the role of T_RM_ cells in long-term viral control. In contrast, T_RM_ cells did not differ in chronic HBV-infected patients regarding their liver damage and viral control status, represented by their specific antigen and antibody response [11]. Liver CD8+ T_RM_ cells are very effective in their function, indicated by secretion of pro-inflammatory cytokines. They also express high levels of IL-2 and perforin, showing their ability for effective direct killing, expansion and proliferation. Their high IL-2 expression further contributes to overcome PD-1L-mediated inhibition and exhaustion, mainly in CD69+CD103+ T_RM_ cells [83]. PD-1L expression is upregulated in liver sinusoidal endothelial cells and hepatocytes upon viral infection. T_RM_ cells highly express the exhaustion markers CD39 and PD1 and interact with PD-1L on intrahepatic cells, which can dampen the pro-inflammatory T_RM_ cell response.

Virus-specific T cells are the key cells for an effective antiviral response in HBV infection. It could be shown that approximately 90% of them have a T_RM_ cell-like phenotype (CD69+CD103+ or CD69+CD103−) [11]. By analyzing an allograft of a donor with HBV infection, it could be demonstrated that donor-derived, virus-specific T_RM_ cells persist for a very long time. In addition, the recipient also develops virus-specific T_RM_ cells that were detectable within the donor-derived liver tissue and interestingly also in the blood, albeit in a lower quantity [10]. In HBV-related HCC, CD8+ T_RM_ cell enrichment could be correlated to an improved prognosis [84]. Thus, T_RM_ cell expansion might be a potential therapeutic target in chronic HBV infection and HCC treatment (Figure 2).

Lymphocytic choriomeningitis virus (LCMV) can cause a systemic acute and chronic infection in mice. During LCMV infection, it could be demonstrated that virus-specific T_RM_ cells in the liver could be influenced by other liver-resident immune cells in order to influence effective pathogen clearance. The PD-1/PD-1L interaction between NK and T_RM_ cells negatively influences T_RM_-dependent cytokine expression in acute and chronic LCMV, and in adenovirus infection. PD-1L inhibition prevented the NK cell-mediated T_RM_ cell inhibition and improved viral clearance [85]. In the course of LCMV infection, γδ T cells contribute to pathogen clearance by producing IFN-γ and TNF-α, migrate and expand at the site of infection [86].

By analyzing CMV-specific T_RM_ cells in human liver allografts, it could interestingly be shown that CMV-specific T cells in humans did not acquire a T_RM_ phenotype in the liver. This phenomenon could be explained by the lack of antigen within the liver [10].

Taken together, the role of T_RM_ cells in viral clearance is not fully elucidated yet.

### 4.2. Liver-Resident T Cells in Parasite Infection

In addition to viral infections, T_RM_ cells play a key role in the infection with parasites. Not many studies exist investigating the role of T_RM_ cells in this context.

*Leishmaniasis* is a parasite induced disease and can be transmitted by a certain group of sandflies. The clinical manifestation of the disease can evolve in a cutaneous, mucocutaneous and visceral manner. Different parasite types exist in which *Leishmania infantum* mainly infects the liver. It can cause acute and resolving hepatic infection, whereas parasite persistence occurs in the spleen. During chronic parasite infection the immune system is not able to fully clear the infection but forms a granuloma to control it. The tissue damage in the liver is relatively limited due to an effective immune response by T_RM_ development. Liver T_RM_ cells are generated and are described to play a protective role during this infection. Thus, it could be shown in different mouse models that circulating memory cells do not provide an effective viral control [87]. Apart from the liver, T_RM_ cells also develop in the skin or other sites of infection. Interestingly, they can also be found far away from the primary site of infection. Liver T_RM_ cells thereby play a special role as their development is especially induced by the *Leishmania* proteins LirCyp1 and LirSOD and the strategy to induce liver T_RM_ cells is used in vaccine development strategies [87]. The importance of T_RM_ cell induction in vaccine development could be shown in HPV strategies. The vaccine led to an induction of effective resident T cells [88]. An amplification of the T_RM_ cell response by TLR activation could not be demonstrated.

More importantly and well-studied is the role of T_RM_ cells during malaria infection. Malaria is caused by *Plasmodium* parasites transmitted by *Anopheles* mosquitos. Plasmodia have a complex life cycle, involving stages in the liver, blood and mosquito. Interestingly, *Plasmodium* infection promotes T_RM_ cell development in the liver [66]. During *Plasmodium* infection T cells first get activated in the spleen and form a memory pool that resides in the liver and builds the front line against invading sporozoites. It could be shown in mouse models that circulating memory cells play a role in parasite control but T_RM_ cells are even more protective and modulate the effective immune response. T_RM_ cells mediate cell cytotoxicity and produce pro-inflammatory cytokines, e.g., IFN-γ and TNF-α. T_RM_ cells were shown to patrol in the liver sinusoids to encounter malaria antigens and display a different and more effective migration pattern than other memory T cells. The importance of T_RM_ cells during malaria could be demonstrated in a *P. knowlesi* infection model, where T_RM_ cell depletion abolished a protective and competent immune response [59].

The development of protective T_RM_ cells in *Plasmodium* infection is used in vaccine development. Thus, a recently published study by Holz et al. used a glycolipid-peptide vaccination. They were able to induce intrahepatic *Plasmodium*-specific memory T cells with a T_RM_ phenotype. A second dose of the vaccination even increased the frequency of these cells in the liver. T_RM_ cells were effective in pathogen clearance and the vaccine protected mice from an infection by *Plasmodium berghei*. The induced T_RM_ cells showed a half-life of 425 days and retained protective function in 90% of the animals up to day 200 [89].

Nevertheless, people who live in areas where malaria is endemic interestingly do not acquire an effective protection against reinfection. Furthermore, the existing vaccines showed insufficient protection and long-term efficiency [90].

To conclude, T_RM_ cells are important during parasite infection and could be helpful in long-term protection in vaccine development (Figure 4).

### 4.3. Liver-Resident T Cells in Chronic Inflammatory Diseases

Chronic liver inflammation can subsequently lead to organ fibrosis, liver failure and HCC development. One of the main causes of liver transplantation in Western countries is non-alcoholic fatty liver disease (NAFLD)-induced end-stage liver disease. NAFLD is considered as a hepatic manifestation of the metabolic syndrome, hypertension and type-II diabetes. In addition, increasing age and disease-associated genetic variants could be shown to be risk factors [91]. The pathomechanism of the disease involves a complex immune response and is not fully understood yet. The involvement of a disrupted T cell—and a more pronounced pro-inflammatory immune response—could be shown in several studies [92,93]. Studies showed that liver-resident T cells, mainly γδ T cells, are involved in NAFLD disease progression. By the production of pro-inflammatory cytokines, such as IL-17A, IFN-γ and TNF-α, they contribute to the pathogenic immune response in NAFLD [94]. One subpopulation of γδ T cells in NAFLD patients expressed CD69, CXCR3 and CXCR6, which are important T_RM_ cell markers. Additionally, they showed a liver-restricted TCR repertoire that supports their liver residency and origin [86]. The direct involvement of T_RM_ cells with an αβ TCR could not be proven yet but it could be possible that they contribute to the overall pro-inflammatory immune response as they are involved in fibrosis in many other organs, e.g., the lung and kidney. Furthermore, it already was shown that the circulating memory T cell quantity is increased in NAFLD patients [95].

There are some indications that the systemic inflammation in obese patients is associated with an increase of T_RM_ cells in the liver, which consequently induces chronic liver inflammation and may be associated with NAFLD disease progression. It was nicely shown in a study from Conroy et al. that there is an increase in liver T_RM_ cells in patients with esophageal adenocarcinoma (OAC). OAC is connected with obesity and the patient cohort had an average BMI of 25.8, but more importantly, an increase in visceral fat. Interestingly, it could be shown that activated T_RM_ cells were increased in the visceral fat and liver tissue of these patients, indicating a systemic immune activation. The activated T_RM_ cells in the liver and adipose tissue produced pro-inflammatory cytokines, such as IL-1β, IL-12, GM-CSF, IL-6, IL-2, IL-4 and IL-15 [96]. These findings indicate that T_RM_ cells could contribute to the overall pro-inflammatory phenotype in obese patients, which is one of the reasons for chronic inflammatory diseases like NAFLD and cancer development (Figure 4).

Extensive transcription factor and gene set enrichment analysis of lung T_RM_ cells showed that, under inflammatory conditions, several drivers of the T cell effector function were overexpressed in these cells, such as RUNX3, IRF4 and NF-kB [97]. These inflammatory markers are known to be upregulated in chronic liver diseases such as ASH and NASH. There, the upregulation of the NF-kB pathway is well described, playing a major role in disease progression [98]. The exact involvement of NF-kB in T_RM_ differentiation and function is not well studied yet, but it is possible that NF-kB signaling is involved as a crucial integrator within the pro-inflammatory cytokine milieu in the liver.

Recent studies could identify the crucial role of changes in the gut microbiota composition and an impairment of the intestinal barrier function on the regulation of body weight and the body’s fat composition. The translocation of bacterial products from the gut into the liver was further shown to contribute to disease progression in NASH or other chronic inflammatory liver diseases [99]. Inflammasomes are important mediators of the innate immune response that get activated upon recognition of pathogen-associated molecular patterns (PAMPs) in the liver. Activation of the NLR family pyrin domain containing 3 (NLRP3) inflammasome and the subsequent release of IL-1ß and IL-18 is implicated in the pathogenesis of chronic inflammatory liver diseases such as ASH and NAFLD [100]. Studies using NLRP3 knockout mice could show that the ablation of inflammasome activation not only prevents from obesity-induced inflammation in fat deposits and the liver but also directly influences the composition of intrahepatic and fat tissue T cells by increasing the amount of naïve T cell numbers and reducing the numbers of effector and memory T cells [101]. Thus, a direct effect of inflammasome activation on T_RM_ cells is plausible but not shown yet.

### 4.4. Liver-Resident T Cells in Cancer

The liver is the primary site for HCC development and is often affected by metastasis of other cancers. HCC can be induced by several factors, e.g., chronic viral infection, alcohol consumption or NAFLD. HCC can interestingly develop in fibrotic and non-fibrotic tissue. In general, effector T cells are important to moderate anti-tumor immunology in HCC development. Furthermore, in several solid tumors and HCC, CD103+ T_RM_ cells are enriched and associated with better prognosis and patient outcomes [84]. In contrast, a recently published study by Williams et al. showed that mice with an altered p21 expression have an increased risk of HCC development. The kinase inhibitor p21 promotes cell cycle arrest and has anti-proliferative functions. In this special situation, T_RM_ cells were expanded in mice with HCC and decreased in line with HCC reduction [102]. Another study showed that T_RM_ cells in HCC have an exhausted phenotype, e.g., shown by the expression of PD-1, LAG-3 and TIM3, especially in comparison to T_RM_ cells, which are not in the tumor microenvironment. During HCC progression, the T_RM_ cell frequency decreased, showing their importance in the anti-tumor immune response. Modern tumor therapies include immunotherapies that targets checkpoint inhibition to promote an anti-tumoral immune response. One important target is PD-1. Thus, during anti-tumor immunotherapy, T_RM_^PD1high^ cells are the most responsive cells to the anti-PD-1 therapy to overcome tumor growth and progression [103]. Furthermore, Ma et al. could demonstrate that T_RM_ cells in a tumor environment express other exhaustion and inhibitory markers, e.g., TIM3 and CTLA-4, but produce pro-inflammatory cytokines such as IFN-γ and TNF-α. Additionally, PD-1 expression in T_RM_ cells within the tumor was correlated to poor disease outcome [104]. Overall, it could be demonstrated that T_RM_ cells and T_RM_ cell function is important for HCC development and anti-tumor therapy (Figure 2).

### 4.5. Liver-Resident T Cells in Transplantation

Solid organ transplantation can be the last therapeutic option in several end-stage organ diseases, such as acute liver failure and cirrhosis. It could be demonstrated that donor-derived T_RM_ cells are detectable in the allografts and that their abundancy could be correlated with organ survival and reduced rejection. Although, organ rejection was associated with increased donor-derived T_RM_ cell abundancy in areas of tissue damage [105,106,107]. A recently published study by Pallett et al. nicely demonstrated that long-lived T_RM_ cells are detectable in liver allografts. Additionally, it could be shown that these cells can migrate into the draining lymph nodes, while downregulating some T_RM_-specific markers like CXCR6; however, they were not measurable intravascular. Furthermore, recipient-derived T_RM_ cells were also abundant in the liver allografts. Interestingly, the same study could demonstrate that donor-derived T_RM_ cells from an HBV-infected donor were still detectable in the liver allograft years after transplantation. Likewise, recipient-derived virus-specific T cells were detectable in the blood and liver. The latter had also a T_RM_-like phenotype. To conclude, this study could generate an important overview of the longevity, plasticity and phenotype of T_RM_ cells in liver transplantation [10]. Further studies are needed to investigate the concrete role of liver T_RM_ cells in organ rejection and function.

## 5. Conclusions

Liver inflammation can be caused by many different diseases but for little of them the importance of T_RM_ cells remains to be investigated. Thus, it is possible that T_RM_ cells play a role in liver autoimmune diseases as these are typically associated with an unbalanced T cell response. T_RM_ cell involvement is already described in multiple sclerosis or type 1 diabetes and are best understood in autoimmune diseases of the skin. In these diseases, T_RM_ cells were shown to be increased and correlated with disease severity. In addition, T_RM_ cell abundancy was associated with pro-inflammatory cytokine expression [108]. A similar mechanism could be possible for autoimmune diseases in liver tissue.

Some studies indicate an involvement of T_RM_ cell development in protective vaccination, but little is known about T_RM_ cell development after vaccination in the liver. The importance of liver T_RM_ cells is only described for parasite infections, as mentioned before. In other organs, T_RM_ development upon vaccination could already be demonstrated. For instance, the commonly used influenza vaccine induces the development of long-lived T_RM_ cell subpopulations independently of the use of neutralizing antibodies or the depletion of circulating memory T cells in mouse experiments. T_RM_ cells were thereby sufficient to protect against reinfection [109].

The function and role of liver T_RM_ cells is a relatively new research field with many unanswered questions. Although there are many interesting new studies, further investigations are urgently needed to analyze the role and importance of these cells in different diseases.

## Figures and Tables

**Figure 1 cells-09-02457-f001:**
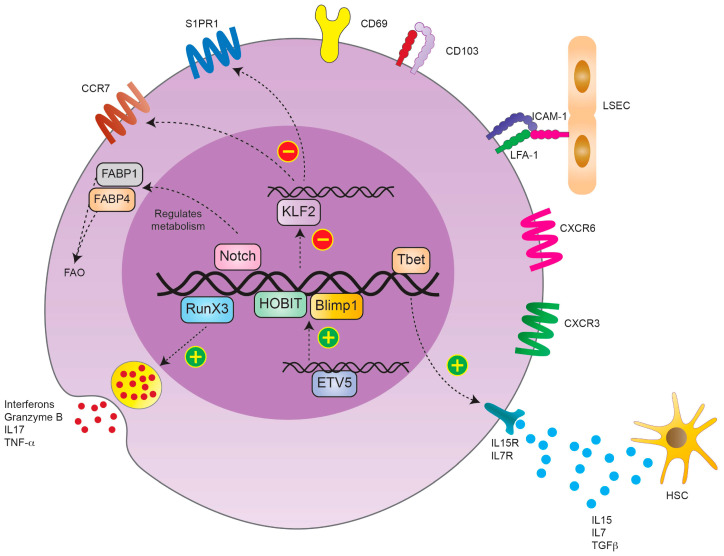
To convey hepatic infiltration, memory T cells express high levels of liver-specific homing markers such as CD103, LFA-1, CXCR6 or CXCR3. Within the liver, T_RM_ cells are exposed to a variety of environmental conditions that directly influence their maintenance, function and development; for example, nutrient deprivation and fatty acid oxidation, hypoxia and different inflammatory conditions. Besides the expression of IL-15, IL-7 and TGF-β, which directly effects hepatic stellate cells (HSCs), intrahepatic T_RM_ cells are also able to secret pro-inflammatory molecules such as Interferon, Granzyme B, IL-17 and TNF-α. Liver-resident T_RM_ cells have the ability to egress back into the bloodstream by upregulating CCR7 and S1PR1.

**Figure 2 cells-09-02457-f002:**
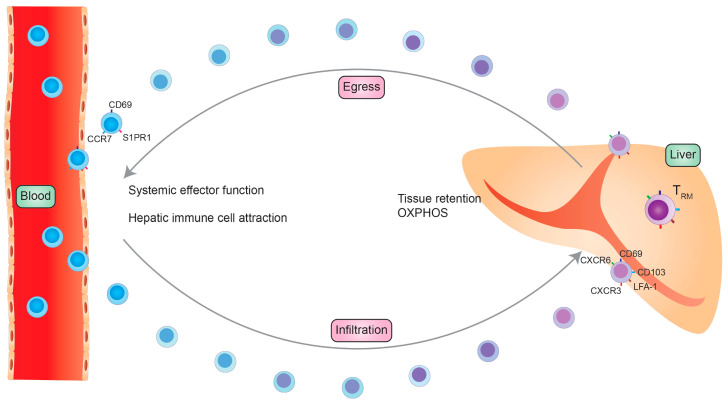
Upon stimulation, T_RM_ cells can recirculate into the blood by inducing the expression of higher levels of CD69, CCR7 and S1PR1. In the blood they can have systemic effector functions and convey hepatic immune cell attraction. For the infiltration into the liver, memory T cells express high levels of the hepatic homing markers LFA-1, CD103, CXCR6, CXCR3 and CD69. In the liver they downregulate the expression of these homing receptors and differentiate into liver-resident T_RM_ cells.

**Figure 3 cells-09-02457-f003:**
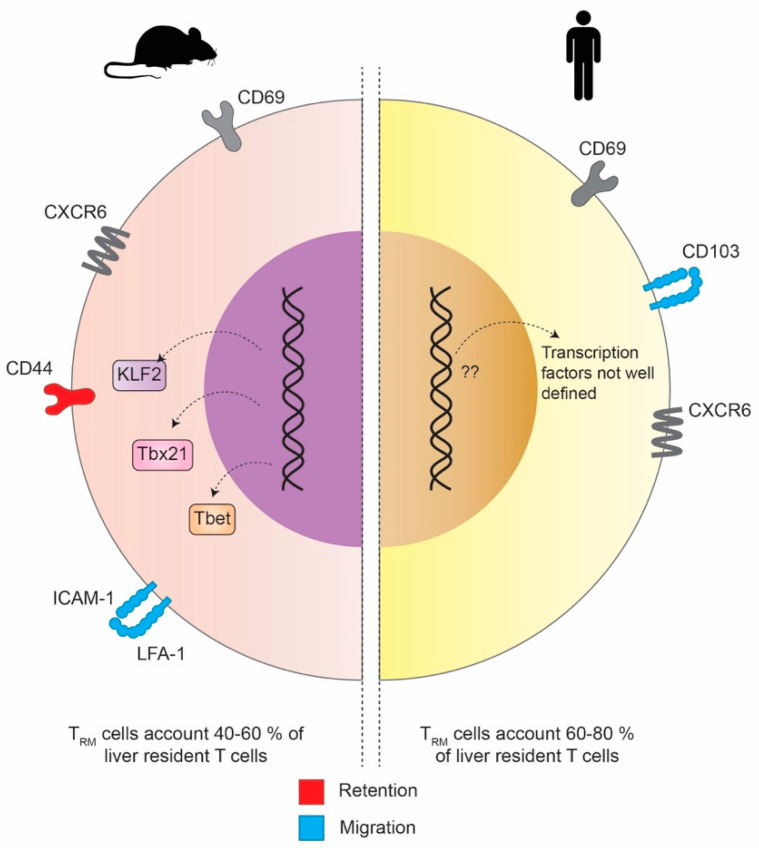
T_RM_ cells can be detected in mice and humans. In both species, T_RM_ cells express CD69 and CXCR3. However, in humans, one population of T_RM_ cells expresses CD103+, while CD103 surface expression is not found in mice T_RM_ cells. On the other hand, mice T_RM_ cells express CD44, which mediates their liver retention and ICAM-1/LFA-1 responsible for T_RM_ cell migration within the liver. The frequency of T_RM_ cells within the resident T cell compartment varies between mice and humans. Some studies have described that mice T_RM_ cells express the transcription factors Tbet, Tbx21 and KLF2, but the transcription factors upregulated by human T_RM_ cells are not defined yet.

**Figure 4 cells-09-02457-f004:**
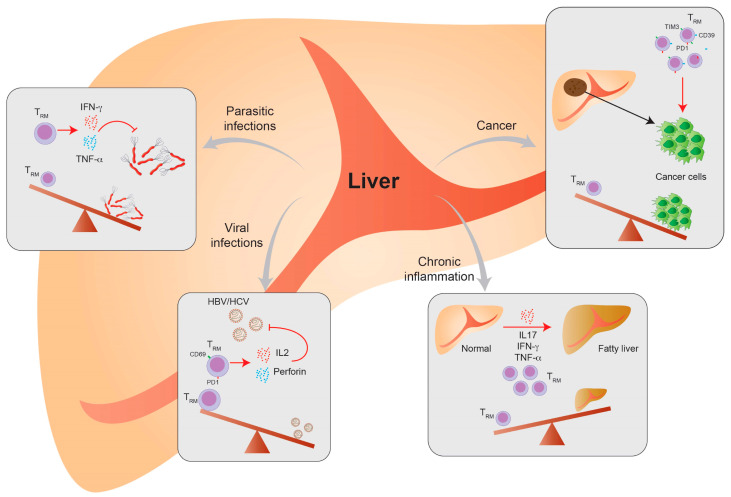
T_RM_ cells play a central role during intrahepatic inflammation. In parasitic infections, T_RM_ cells mediate cytotoxicity by the production of pro-inflammatory cytokines such as IFN-γ and TNF-α. During hepatitis virus infection, virus-specific T_RM_ cells could be negatively correlated with virus titers by secreting high levels of IL-2 and perforin. Patients who attained viral control showed enriched intrahepatic T_RM_ cell numbers. In sterile chronic inflammation of the liver, such as alcoholic or nonalcoholic steatohepatitis (ASH and NASH) T_RM_ cells were associated with the overall pro-inflammatory cytokine milieu in the liver. Their expression of IL-17, IFN-γ and TNF-α contributes to disease progression. During the subsequent development of hepatocellular carcinoma (HCC), the CD103+ T_RM_ cells showed opposing effects. Patients with enriched numbers of these cells showed a better prognosis by mediating the anti-tumoral immune response.

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
