# Peer review of "Tissue-Resident Memory T Cells in the Liver—Unique Characteristics of Local Specialists"

_cells, 2020, doi:10.3390/cells9112457_

Round 1

Reviewer 1 Report

Bartsch et al. extensively reviewed that the characteristics of TRM cells in the liver. This review seems important in this area.

  1. Please give your comments about the association among natural killer T (NKT) cells, NK cells and TRM cells in the liver. The Hobit-Blimp1 transcriptional module is also required for other populations of tissue-resident lymphocytes, including NKT cells and liver-resident NK cells, all of which share a common transcriptional program [18]. Hobit and Blimp1 as central regulators of this universal program that instructs tissue retention in diverse tissue-resident lymphocyte populations [18]. Short-term in vivo blockade of the ARTC2.2/P2RX7 axis permits much improved flow cytometry-based phenotyping and enumeration of murine peripheral invariant NKT cells (iNKT) and TRM from nonlymphoid tissues, and it represents a crucial step for functional studies of these populations [Borges da Silva H et al. J Immunol. 2019 Apr 1;202(7):2153-2163.].
  2. Please give your comments about the interaction between TRM cells and TLR signaling. See: Marinaik CB, et al. Cell Rep Med. 2020 Sep 22;1(6):100095.; Thompson EA, et al. Cell Rep. 2019 Jul 30;28(5):1127-1135.e4. doi: 10.1016/j.celrep.2019.06.087.; Çuburu N, et al. J Immunol. 2019 Feb 15;202(4):1250-1264. doi: 10.4049/jimmunol. It was reported that modulation of TLR signaling pathway may improve the treatment of HCC patients with refractory disease (Sasaki R, et al. Int J Mol Sci. 2020 May 9;21(9):3349. doi: 10.3390/ijms21093349.).

Author Response

Response to Reviewer 1:

Bartsch et al. extensively reviewed that the characteristics of TRM cells in the liver. This review seems important in this area.

1. Please give your comments about the association among natural killer T (NKT) cells, NK cells and TRM cells in the liver. The Hobit-Blimp1 transcriptional module is also required for other populations of tissue-resident lymphocytes, including NKT cells and liver-resident NK cells, all of which share a common transcriptional program [18]. Hobit and Blimp1 as central regulators of this universal program that instructs tissue retention in diverse tissue-resident lymphocyte populations [18]. Short-term in vivo blockade of the ARTC2.2/P2RX7 axis permits much improved flow cytometry-based phenotyping and enumeration of murine peripheral invariant NKT cells (iNKT) and TRM from nonlymphoid tissues, and it represents a crucial step for functional studies of these populations [Borges da Silva H et al. J Immunol. 2019 Apr 1;202(7):2153-2163.].

Response to 1:

We want to thank the reviewer for the valuable comments. We agree that the association among natural killer T (NKT) cells, NK cells and TRM cells in the liver is important and added a paragraph related to this topic to ‘4. Liver TRM cells – in health, disease and vaccination’ lines 421-433. This paragraph reads:

‘In addition, Natural killer cells (NK cells) and NKT cells contribute to the pro-inflammatory immune response against pathogens and against cancer cells within the liver. NKT cells are a cell population which combines T cell and NK-cell markers and functions. NK cell and NKT cell subsets are enriched in liver sinusoids and complement the immune surveillances function against pathogens and toxins which reach the liver mainly through the portal vein from the gut [76]. Interestingly, liver resident NK-and NKT cells display a higher function and cytotoxicity in comparison to their circulating counterparts [77]. Similar to TRM cells, NK- and NKT cells play a role in chronic liver disease, such as liver fibrosis, hepatocellular carcinoma (HCC) and viral hepatitis. In some diseases they complement TRM cell function, whereas in others they have an opposing function. One challenge is to analyze NK-T cells in non-lymphoid tissues. Inhibition of P2RX7 signaling could be a strategy to restore their function after tissue isolation in order to quantify their frequency in healthy and infected liver tissue [78]. By using this strategy, the investigation of the importance and function of NK-T cells could be improved.' 

2. Please give your comments about the interaction between TRM cells and TLR signaling. See: Marinaik CB, et al. Cell Rep Med. 2020 Sep 22;1(6):100095.; Thompson EA, et al. Cell Rep. 2019 Jul 30;28(5):1127-1135.e4. doi: 10.1016/j.celrep.2019.06.087.; Çuburu N, et al. J Immunol. 2019 Feb 15;202(4):1250-1264. doi: 10.4049/jimmunol. It was reported that modulation of TLR signaling pathway may improve the treatment of HCC patients with refractory disease (Sasaki R, et al. Int J Mol Sci. 2020 May 9;21(9):3349. doi: 10.3390/ijms21093349.).

Response to 2:

We agree that the interaction between TRM cells and TLR signaling is important to be mentioned. In response to the reviewers comment we added a paragraph addressing this to 4. Liver TRM cells – in health, disease and vaccination’ lines 434-444. This paragraph now reads:

The interaction of the innate and adaptive immune system in various diseases is tremendously important. It could be shown that modern HCC treatment strategies upregulate Toll like receptor (TLR) signaling pathways [79]. This general immune activation could be the link to TRM cell development in this treatment strategy by the secretion of cytokines such as IL-10 [29]. In vaccine studies in the lung against influenza viruses, adjuvants which are TLR agonist induce an effective polyfunctional T cell immunity, nevertheless the induction of CD69 and CD103 expression was similar between different adjuvants [80]. A direct activation and differentiation of TRM cells by TLR activation could not be shown in the liver yet. It is known that TLR activation occurs in the liver subsequently to the leaky gut syndrome in several chronic inflammatory diseases. Therefore, the investigation of alterations in TRM differentiation and function by TLR signaling is a very important
topic for future investigations.'

In addition, we added a statement to point ‘4.2. Liver resident T cells in parasite infection lines 513-515.:

‘The importance of TRM cell induction in vaccine development could be shown in HPV strategies. The vaccine leaded to an induction of effective resident T cells [88]. An amplification of the TRM cell response by TLR activation could not be demonstrated.

Reviewer 2 Report

Title: Tissue-Resident Memory T Cells in the Liver – Unique Characteristics of Local Specialists.

The authors reviewed the role of tissue-resident memory T cells (TRM cells) in the liver. They describe that TRM cells are important for the immunological role of the liver. The liver is exposed to several toxins and pathogens, including many viruses, therefore TRM cells play a pivotal immunogenic role in this organ. There are no similar reviews published, and in my opinion this one is important because it highlights the importance of these cells and illuminates the pathological molecular mechanism of liver diseases, such as NASH, viral hepatitis and hepatocellular carcinoma where TRM cells are involved. Some therapeutic targets can be found in this review.

This is an original, novel, updated and well-written review, that in my opinion, deserves publication in Cells. However, I have some comments:

1.- Perhaps two more figures will help to make this review more didactical.

2.- Authors may add a section dealing with the importance of gut dysbiosis and TMR cells in the contest of liver diseases.

3.- This review can be improved if authors discuss the role of NF-kappaB and NLRP3 signaling pathways on TMR cells during liver inflammation.

4.- Do free radicals play an important role in TMR cell activity?

Author Response

Response to Reviewer 2:

The authors reviewed the role of tissue-resident memory T cells (TRM cells) in the liver. They describe that TRM cells are important for the immunological role of the liver. The liver is exposed to several toxins and pathogens, including many viruses, therefore TRM cells play a pivotal immunogenic role in this organ. There are no similar reviews published, and in my opinion this one is important because it highlights the importance of these cells and illuminates the pathological molecular mechanism of liver diseases, such as NASH, viral hepatitis and hepatocellular carcinoma where TRM cells are involved. Some therapeutic targets can be found in this review.

This is an original, novel, updated and well-written review, that in my opinion, deserves publication in Cells. However, I have some comments:

1.- Perhaps two more figures will help to make this review more didactical.

Response to 1:

We want to thank the reviewer for his/her very helpful and valuable comments. We agree with the reviewer that two more figures help to make the review more didactical. Therefore, we split Figure 1 and created a new Figure 2. I addition, we added a new Figure 3 to point out the known differences in the phenotype of TRM cells between mouse and men.  

2.- Authors may add a section dealing with the importance of gut dysbiosis and TMR cells in the contest of liver diseases.

3.- This review can be improved if authors discuss the role of NF-kappaB and NLRP3 signaling pathways on TMR cells during liver inflammation.

Response to 2/3:

We agree with the reviewer that addressing the importance of the influence of gut dysbiosis, NLRP3 signaling and NF-kappaB activation on TRM cell differentiation and function in the context of liver diseases is very important and added a paragraph to point ‘4.3 Liver resident T cells in chronic inflammatory diseases’ lines 569-588.  This new paragraph now reads:

‘Extensive transcription factor and gene set-enrichment analysis of lung TRM cells showed that under inflammatory conditions, several drivers of T cell effector function were overexpressed in these cells such as RUNX3, IRF4 and NF-kB [98]. These inflammatory markers are known to be upregulated in chronic liver diseases such as ASH and NASH. There, the upregulation of the NF-kB pathway is well described to play a major role in disease progression [99]. The exact involvement of NF-kB in TRMdifferentiation and function is not well studied yet, but it is possible that NF-kB signaling is involvedas crucial integrator within the pro-inflammatory cytokine milieu in the liver. Recent studies could identify a crucial role of changes in the gut microbiota composition and an impairment of the intestinal barrier function on the regulation of body weight and the bodies fat composition. The translocation of bacterial products from the gut into the liver was further shown to contribute to disease progression in NASH or other chronic inflammatory liver diseases [100]. Inflammasomes are important mediators of the innate immune response that get activated upon recognition of pathogen-associated molecular patterns (PAMPs) in the liver. Activation of the NLR family pyrin domain containing 3 (NLRP3) inflammasome and the subsequent release of IL-1ß and IL-18 is implicated in the pathogenesis of chronic inflammatory liver diseases such as ASH and NAFLD [101]. Studies using NLRP3 knockout mice could show, that the ablation of inflammasome activation not only prevents from obesity-induced inflammation in fat deposits and the liver but also directly influences the composition of intrahepatic and fat tissue T cells by increasing the amount of naïve T cell numbers and reducing the numbers of effector and memory T cells [102]. Thus, a direct effect of inflammasome activation on TRM cells is plausible but not shown yet.’

4.- Do free radicals play an important role in TMR cell activity?

We want to thank the reviewer for pointing out this interesting question. To address this, we added a paragraph to point ‘2.2.2 Factors influencing/driving TRM cell development’ lines 191-194. This paragraph reads:

 ‘Another cytokine which functions as an autocrine stimulus on CD8 TRM cells is IL-2. Pallett et al. showed that human liver CD8 TRM cells produce high levels of interleukin-2 (IL-2), which was shown to be critical for intrahepatic TRM cell survival, ability to function and for their antigen specific proliferation [11]. Increased IL-2 expression could also be shown in response to oxidative stress induced by stimulation with reactive oxygen species and free radicals which is a common characteristic of chronic liver diseases such as non-alcoholic steatohepatitis (NASH) [31]. Thus, oxidative stress induction could alter TRM cell function and differentiation.’
